# Meis1 Controls the Differentiation of Eye Progenitor Cells and the Formation of Posterior Poles during Planarian Regeneration

**DOI:** 10.3390/ijms24043505

**Published:** 2023-02-09

**Authors:** Shaocong Wang, Yujia Sun, Xiaomai Liu, Yajun Guo, Yongding Huang, Shoutao Zhang, Qingnan Tian

**Affiliations:** 1School of Life Sciences, Zhengzhou University, Zhengzhou 450001, China; 2Longhu Laboratory of Advanced Immunology, Zhengzhou 450046, China

**Keywords:** *DjMeis1*, *Djwnt1*, planarian, regeneration, stem cells, Wnt

## Abstract

As a member of TALE family, Meis1 has been proven to regulate cell proliferation and differentiation during cell fate commitment; however, the mechanism is still not fully understood. The planarian, which has an abundance of stem cells (neoblasts) responsible for regenerating any organ after injury, is an ideal model for studying the mechanisms of tissue identity determination. Here, we characterized a planarian homolog of Meis1 from the planarian *Dugesia japonica*. Importantly, we found that knockdown of *DjMeis1* inhibits the differentiation of neoblasts into eye progenitor cells and results in an eyeless phenotype with normal central nervous system. Furthermore, we observed that *DjMeis1* is required for the activation of Wnt signaling pathway by promoting the *Djwnt1* expression during posterior regeneration. The silencing of *DjMeis1* suppresses the expression of *Djwnt1* and results in the inability to reconstruct posterior poles. In general, our findings indicated that *DjMeis1* acts as a trigger for the activation of eye and tail regeneration by regulating the differentiation of eye progenitor cells and the formation of posterior poles, respectively.

## 1. Introduction

Meis1 protein is a member of the three amino acid loop extension (TALE) transcription factors family. Previous studies have demonstrated that Meis1 plays essential roles in the embryonic development across metazoans. For example, Meis1 participates in the embryonic cortical development by promoting neuronal proliferation and migration [1]. Cooperating with Tbx and Hox factors, Meis1 acts as a trigger for limb initiation. Knockdown of *Meis1* results in severe limb agenesis [2]. In addition, Meis1 is involved in organ patterning, including the formation of hearts [3], eyes, and lungs [4,5]. With the exception of embryonic development, biological functions of Meis1 have a close correlation with the tumorigenesis [6,7]. Upregulation of *Meis1* and associated co-factors can promote the skin tumorigenesis. Knockdown of *Meis1* induces a significant decrease in benign and malignant tumors in mice [8]. Meanwhile, Meis1 also serves as a negative regulator in the occurrence and development of cancers, such as non-small cell lung cancer and prostate cancer [9,10]. In general, these studies suggest that Meis1 has broad functions in the regulation of cell proliferation and differentiation during embryonic development. However, the mechanism of Meis1 family, which is responsible for stem cells proliferation and differentiation, remains unclear.

Planarians can regenerate a complete individual from any fragment of their body [11,12,13]. This strong regenerative ability depends on an abundance of stem cells (neoblasts) that exist in the adult body [14,15,16,17,18]. The regeneration of heads or tails in planarians, known as anterior-posterior (AP) axis regeneration, is an important model for studying the mechanism of stem cells proliferation, differentiation, migration, and apoptosis [19,20]. The position control genes (PCGs) are genes with a regional expression pattern and are constitutively expressed in muscles. By encoding positional information, PCGs determine the cell fate along the anterior-posterior axis, medial-lateral axis, and dorsal-ventral axis. Knockdown of PCGs will lead to mis-patterning of the phenotypes [19]. At present, Wnt signaling is known to regulate the tissue identity along the AP axis during regeneration [21,22,23]. Wnt signaling pathway components, *Djwnt1* and *β-catenin,* act as promoters for the construction of posterior poles [24,25]. Knockdown of *Djwnt1* can cause planarians that are unable to regenerate tails or regenerate heads in the place of tails. *β-catenin* acts as a downstream gene of *Djwnt1.* Silencing of *β-catenin* can result in the reversal of posterior poles and cause planarians to regenerate posterior heads from posterior blastema [26]. Furthermore, *notum* serves as an inhibitor of *Djwnt1* during anterior poles regeneration [27]. *Notum* RNAi can upregulate Wnt signaling and cause animals to regenerate tails in the place of heads [28]. Recently, *Djpbx* and *Djislet* have been proven to be positive regulators of *Djwnt1* during posterior poles regeneration. Knockdown of *Djpbx* and *Djislet* can cause planarians that are unable to regenerate their tails [29,30]. In general, these studies suggest that PCGs involved in the Wnt signaling pathway act as a determinant of the cell fate along the AP axis. However, the underlying mechanisms in regulating the Wnt signaling are not yet fully understood.

In our work, we identify a homolog of *Meis1* in planarian *Dugesia japonica.* We observe that *DjMeis1* RNAi planarians cannot form mature eye cells or reconstruct the posterior poles. *DjMeis1* RNAi decreases the number of eye progenitor cells and suppresses the expression of *Djwnt1*. Herein, we propose that *DjMeis1* serves as a positive factor in the regulation of the differentiation of eye progenitor cells and the establishment of posterior poles during planarian regeneration.

## 2. Results

### 2.1. DjMeis1 RNAi Inhibits the Eye and Tail Regeneration

To better investigate the role of Meis1 in the tissue identity determination, we identified three homologous proteins of Meis in planarian *Dugesia japonica* based on our previous transcriptome [31]. Two of the Meis homologous *DjMeis2* and *DjMeis3* have been previously reported as *SmedMeis* and *SmedMeis-like* in planarian *Schmidtea mediterranea* (Appendix A) [32,33]. RNAi experiments were performed by injecting dsRNA to the planarians. At 24 h after the last injection, planarians were amputated from the anterior and posterior sites of pharynx into three fragments: Head, trunk, and tail fragments (Figure 1A). We observed that the fragments of control groups regenerated into complete individuals at 7 days after amputation (7 dpa) (Figure 1B). Consistent with the phenotypes previously reported, silencing of *DjMeis2* in *Dugesia japonica* resulted in the formation of smaller eyes (Figure 1C). *DjMeis3* RNAi induced the formation of a squared head with elongated eyes or cyclops (Figure 1D and Appendix A). Importantly, we identified a rare eyeless combined with tailless phenotype (Figure 1E), due to the inhibition of *Meis1*, a *Meis1* homolog that we named *DjMeis1*. The *DjMeis1* RNAi eyeless phenotype was reminiscent of the phenotypes observed from *DjMeis2* and *DjMeis3* RNAi animals, which suggested a conserved role for Meis in the eye development. In contrast to the *DjMeis2* and *DjMeis3* RNAi phenotypes, *DjMeis1* RNAi animals displayed more sever inhibition in the eye regeneration. No eyes could be observed in newly regenerated heads in *DjMeis1* RNAi animals. Furthermore, head fragments of *DjMeis1* RNAi animals failed to regenerate tails (Figure 1E). Trunk fragments could regenerate eyeless heads from anterior blastema, but were unable to develop tails from posterior blastema (Figure 1E). These phenotypes indicate that *DjMeis1* may have broad requirements in the eye and tail regeneration.

### 2.2. DjMeis1 Is Required for the Eye Regeneration by Regulating the Formation of Eye Progenitor Cells

It has been proven that PCGs expressed in muscles determine the tissue identity, including the specification of neoblasts and the correct location of progenitor cells [19,34,35]. To verify whether the eyeless phenotype was due to the effect of *DjMeis1* RNAi on body regionalization, we detected the expression of *ndl2*, a PCG that was expressed in the pre-pharyngeal regions and defined the head patterning [36,37]. Compared to the control animals, *DjMeis1* RNAi animals displayed a normal expression pattern of *ndl2*, suggesting that *DjMeis1* RNAi might not impair the expression of anterior PCGs (Figure 2A).

The brain nerves in planarian could be formed within 24 h after amputation and continued to develop into a complete bilobed brain. The first photoreceptor neurons regenerated in the dorsal side of the brain within 3 days [38]. In the view of the close connection between the development of eyes and brain nerves in planarians, we speculated that knockdown of *DjMeis1* might impair the reconstruction of brain nerves, and then lead to the failure of eye regeneration. To test this possibility, we performed WISH with *sert* (*sert* is expressed in serotonergic neurons) and *PC2* (*PC2* is expressed in the central nervous system (CNS)) probes to detect the reconstruction of brain nerves [39,40,41]. We observed that the bilobed brain of *DjMeis1* RNAi animals was normally reconstructed compared to the control animals at 7 dpa (Figure 2B,C), suggesting that knockdown of *DjMeis1* had no effect on the regeneration of brain nerves, and *DjMeis1* acted independently of the brain nerves in the regulation of eye development.

To clarify the effect of *DjMeis1* RNAi on the formation of eyes, we performed WISH with *Djovo* probe (*Djovo* is expressed in the eye progenitor cells) to examine the regeneration of eye progenitor cells in *DjMeis1* RNAi animals [42]. Compared to the control animals, the number of *Djovo^+^* cells in *DjMeis1* RNAi animals was significantly reduced (Figure 2D), suggesting that the deletion of *DjMeis1* suppressed the differentiation of eye progenitor cells from neoblasts, and *DjMeis1* was required for the formation of eye progenitor cells. Previous studies on the development of the visual system demonstrated that *Djsix1/2* and *Djeya* were also essential for the regeneration of eye progenitor cells, and these specific transcription factors were co-expressed with *Djovo* [32,43,44]. To further explore the molecular functions of *DjMeis1* on the development of eye progenitor cells, we detected the expression pattern of *DjMeis1* after knockdown of *Djovo*, *Djsix1/2,* and *Djeya*, respectively. We found that planarians with the silencing of *Djovo*, *Djeya,* and *Djsix1/2* greatly reduced the *DjMeis1* signal compared to the control groups (Appendix A). These results suggest that *DjMeis1* is required for the differentiation of eye progenitor cells expressing *Djovo*, and in turn, the formation of eye progenitor cells promote the expression of *DjMeis1*. Furthermore, to examine whether the differentiation from eye progenitor cells into mature eye cells was also affected after knockdown of *DjMeis1*, we detected the formation of mature eye cells by WISH for *opsin* [41,45]. We observed that no mature eye cells were regenerated in *DjMeis1* RNAi animals at 7 dpa (Figure 2E), indicating that *DjMeis1* also played an important role in the formation of mature eye cells from eye progenitor cells. To test this model, we performed eye resection to planarians, which retained the pre-existing brain nerves. We observed that *DjMeis1* RNAi animals failed to regenerate their eyes at 7 days after surgical removal (Figure 2F). Considered together, we conclude that *DjMeis1* acts as a trigger for the differentiation of eye progenitor cells and promotes the formation of mature cells without affecting the regeneration of brain nerves.

### 2.3. DjMeis1 Control of the Tail Regeneration Is Independent of Cell Proliferation

Given the requirement of *DjMeis1* in tail regeneration, we first detected the epidermal cells of planarian with *LaminB* probe (the epidermal boundary marker) to determine whether the posterior wound in *DjMeis1* RNAi animals could be healed [46]. We observed that *DjMeis1* RNAi animals showed a normal expression pattern of *LaminB* compared to the control groups at 7 dpa (Figure 3A), suggesting that *DjMeis1* RNAi animals retained the ability to heal the posterior wound despite failing to regenerate their tails.

As planarians remodeling their missing tissues rely on the cell resources provided by the continuous proliferation of neoblasts [16,19], next, we attempted to analyze whether the tailless phenotype caused by *DjMeis1* RNAi was due to the effect on cell proliferation. It has been reported that there are two waves of proliferative response of neoblasts after amputation. The first wave commences at 6 h after injury and shows an increase in the cell proliferation throughout the body. The second wave commences at 48 h and tends to be restricted to the wound site [47]. Therefore, we performed whole-mount immunofluorescence for phosphorylated histone H3 (H3p) and bromodeoxyuridine (BrdU) labeling experiments, and quantified the mitotic density at 6 and 48 h to determine whether the proliferative ability of neoblasts was impaired after *DjMeis1* RNAi [48,49]. However, we observed that *DjMeis1* RNAi animals showed normal proliferative ability compared to the control groups at both 6 and 48 h after amputation (Figure 3B–E). In conclusion, *DjMeis1* RNAi has no effect on the cell proliferation, and the failure of tail regeneration is not caused by the downregulation of the proliferative ability of the neoblasts.

### 2.4. DjMeis1 Is Required for the Re-Establishment of Posterior Poles by Regulating the Expression of Djwnt1

Previous studies have demonstrated that signals of posterior poles are required for tail regeneration [29,30]. Therefore, we speculated that *DjMeis1* RNAi might induce the tailless phenotype by affecting the reconstruction of posterior poles. To test this possibility, we performed whole-mount immunostaining and fluorescence in situ hybridization experiments to detect the ventral nerve cords (VNC) reconstruction. We observed that *DjMeis1* RNAi animals failed to regenerate VNC posteriorly, which was consistent with the result that *DjMeis1* RNAi animals healed the posterior wound at the amputated site without activating the regeneration of tails (Figure 4A and Appendix A) [50,51]. Due to the defects on the regeneration of VNC, we were interested in whether the regeneration of the posterior organ was affected following *DjMeis1* RNAi. Therefore, we detected the regeneration of pharynx (*mhc-1*^+^ cells) of the head fragments and observed that *DjMeis1* RNAi animals regenerated incomplete pharynx or failed to regenerate pharynx (Figure 4B) [37,52], suggesting that *DjMeis1* RNAi inhibited the proper formation of organs during posterior regeneration. Moreover, we performed in situ detection on *fz4* [29], a posterior poles marker of planarian. We found that *DjMeis1* RNAi animals failed to express *fz4* in posterior blastema (Figure 4C). In general, these data suggest that *DjMeis1* acts as an important regulator in the reconstruction of posterior poles.

During planarian regeneration, Wnt signaling is known to determine the posterior poles reconstruction [24]. Knockdown of *Djwnt1* resulted in the inability of planarians to regenerate their tails or to regenerate posterior heads in the place of tails after amputation. Next, we attempted to examine whether the silencing of *DjMeis1* impaired the reconstruction of posterior poles by affecting the expression of *Djwnt1*. It has been reported that there are two phases of *Djwnt1* expression during regeneration. The first phase is detected in the wound site at 24 h after amputation, and then the second phase is observed in the posterior blastema tip at 96 h after amputation [30]. Therefore, we detected the expression pattern of *Djwnt1* at 24 and 96 h after amputation, respectively. We observed that *DjMeis1* RNAi significantly decreased the expression of *Djwnt1* compared to the control groups at 24 h after amputation and completely inhibited the expression of *Djwnt1* at 96 h after amputation (Figure 4D). These results suggest that *DjMeis1* plays an essential role in the expression of *Djwnt1,* which is required for the re-establishment of posterior poles. To test this model, we performed in situ detection on the *Djwnt1* downstream factor *Djwnt11-2* [29,30]. As expected, we observed that the expression of *Djwnt11-2* was lost after knockdown of *DjMeis1* (Figure 4E). From these data, we conclude that *DjMeis1* is an important factor in posterior fate specification and determines the re-establishment of posterior poles by triggering the expression of *Djwnt1*.

### 2.5. β-Catenin/DjMeis1 RNAi Induces the Posterior Poles of Planarians Reversal

*β-catenin* is previously reported as a down-stream gene of *Djwnt1* and is required for the reconstruction of posterior poles. *β-catenin* RNAi animals displayed anteriorization of posterior poles and regenerated heads from posterior blastema [25,53]. In the view of the fact that tailless animals caused by *DjMeis1* RNAi retained the proliferative ability of neoblasts, we were interested in whether the silencing of *β-catenin* could still cause the reversal of the posterior poles of *DjMeis1* RNAi animals. To verify these effects, we performed *β-catenin* and *DjMeis1* double RNAi and observed that double RNAi animals regenerated eyeless heads in the place of tails (Figure 5A). This phenotype confirms that *DjMeis1* functions to promote the reconstruction of posterior poles independently of cell proliferation.

To determine the reversal of posterior poles of double RNAi animals, we detected the expression of *sFRP1* (*sFRP1* is an anterior poles marker of planarian) at 7 dpa and found that *sFRP1*^+^ cells existed both in anterior and posterior poles in double RNAi animals (Figure 5B) [37], suggesting that the event of posterior poles reversal also occurred in double RNAi animals. Since the posterior head of double RNAi animals could not regenerate eyes, we proposed that this eyeless phenotype was caused by the loss of function of *DjMeis1,* which acts on the eye development. Therefore, we detected the reconstruction of CNS and observed that double RNAi animals regenerated another group of brain nerves in the posterior head similar to the *β-catenin* RNAi animals (Figure 5C). Meanwhile, we performed in situ detection on *opsin* and found that the newly regenerated head from posterior blastema of *β-catenin* RNAi animals displayed a clear *opsin* expression pattern. In contrast, no *opsin*^+^ cells were detected in the posterior head of double RNAi animals (Figure 5D). These data suggest that *DjMeis1* has a conservative function on the formation of eyes, even in the ectopic heads. In general, our data prove that *β-catenin* and *DjMeis1* double RNAi can cause the reversal of posterior poles of *DjMeis1* RNAi animals without affecting the proliferation of neoblasts, but they retain the requirement of *DjMeis1* for eye regeneration.

## 3. Discussion

Meis1 is an important transcription factor that is initially discovered in leukemic mice and its biological functions have been extensively studied in leukemia, organogenesis, embryonic development, and tumorigenesis [54]. Recently, Meis1 has been found to be involved in the cell cycle regulation of cardiomyocytes and endothelial cells [4,55]. Based on the previous studies, Meis1 plays an important role in cell differentiation during cell fate commitment, although the mechanism remains unclear. In this study, using the planarian as a model for regeneration, we found that *DjMeis1* acts as a trigger for the activation of eye and tail regeneration by regulating the differentiation of eye progenitor cells and the formation of posterior poles, respectively.

Previous studies in mice demonstrated that Meis1 played essential roles in the eye development, as smaller eye lenses were observed in *Meis1* mutant embryos [5]. Consistently, in this study, we found that *DjMeis1* RNAi planarians exhibited an eyeless phenotype (Figure 1E). It has been reported that there is a close correlation between the formation of eyes and brain nerves in planarians [38]. However, we found that *DjMeis1* RNAi planarians regenerated normal brain nerves compared to the control groups (Figure 2C), suggesting that *DjMeis1* did not play an important role in the reconstruction of brain nerves, and promoted the formation of eyes independently of brain nerves (Figure 6A). By detecting different eye cells lineage during the eye development [41,42], we observed that *DjMeis1* RNAi clearly reduced the number of *Djovo^+^* eye progenitor cells and inhibited the formation of mature eye cells, which differentiated from the early progenitor lineage (Figure 2D,E). Meanwhile, the expression of *DjMeis1* was suppressed after knockdown of *Djovo*, *Djsix1/2,* and *Djeya,* which are essential factors for the regeneration of eye progenitor cells (Appendix A), suggesting that *DjMeis1* participated in the regulation of eye progenitor cells differentiation, and in turn, the formation of eye progenitor cells promoted the *DjMeis1* expression. Furthermore, we observed that no eyes could be formed in *DjMeis1* planarians at 7 days after eye resection (Figure 2F). Considered together, we conclude that *DjMeis1* is required for the differentiation of eye progenitor cells by promoting the *Djovo* expression and promotes the proper maturation of eye cells in planarians (Figure 6B).

Tailless is another phenotype caused by *DjMesi1* RNAi in planarian regeneration (Figure 1E). We envision at least two possible mechanisms by which *DjMeis1* RNAi inhibits the tail regeneration. *DjMeis1* could act as a promoter of cell proliferation or alternatively be required for neoblasts differentiation. It has been reported that Meis1 is involved in the proliferation and migration of pulmonary artery smooth muscle cells (PASMCs) in hypoxia [56]. Recently, Meis1 has been proven to act as a transcription factor to promote hair matrix cell proliferation [57], suggesting that Meis1 has a unique function on the regulation of cell proliferation. However, in this study, we found that *DjMeis1* RNAi planarians displayed normal mitotic activities compared to the control groups (Figure 3B–E), indicating that knockdown of *DjMeis1* had no effect on the proliferation of neoblasts. Previous studies have reported that Meis1 is involved in several steps of limb AP pre-patterning and the elimination of *Meis1* leads to severe standstill of limb development [2]. Next, we tested the possibility that *DjMeis1* might regulate the differentiation of neoblasts by promoting the reconstruction of posterior poles. Indeed, we found that *DjMeis1* RNAi planarians could not properly regenerate VNC and pharynx during posterior reconstruction and failed to express the posterior poles marker *fz4* (Figure 4A–C), suggesting that *DjMeis1* acted as a regulator for the proper reconstruction of posterior poles. Furthermore, the expression of the previously known and important regulator of posterior regeneration, *Djwnt1,* was significantly suppressed at both 24 and 96 h after amputation following the knockdown of *DjMeis1* (Figure 4D). The expression of *Djwnt11-2,* which acted as a downstream gene of *Djwnt1,* was also lost in *DjMeis1* RNAi planarians (Figure 4E and Figure 6C). Considered together, we propose that *DjMeis1* determines the specification of posterior fate by triggering the expression of *Djwnt1* (Figure 6D). In support of this hypothesis, we performed *β-catenin* and *DjMeis1* double RNAi. Knockdown of *β-catenin* could result in the reversal of posterior poles and cause planarians to regenerate posterior heads [25]. In our study, we observed that *β-catenin* and *DjMeis1* double RNAi induced an eyeless head from posterior blastema (Figure 5A), which supported our findings that *DjMeis1* RNAi had no effect on the proliferation of neoblasts, but inhibited the expression of *Djwnt1*. In addition, we observed that double RNAi planarians normally regenerated a pair of brain nerves in the posterior head, but failed to produce mature eye cells (Figure 5C,D), which further confirmed the conserved role of *DjMeis*1 in the differentiation of eye progenitor cells. Another TALE transcription factor, *Djpbx,* has been previously reported to be required for the establishment of anterior-posterior axis [29]. Knockdown of *Djpbx* caused the failure of planarians to regenerate their tails or heads. In contrast to *Djpbx*, which is required for both anterior and posterior patterning along the body axis, *DjMeis1* mainly regulates the posterior patterning. However, both *Djpbx* and *DjMeis1* function as regulators that interpret signals along the AP axis, suggesting that TALE transcription factors have a conservative function along the AP axis.

In conclusion, our findings generally support the deep evolutionary functional conservation of Meis1 in the eye development and re-establishment of body patterning. Our work suggests that *DjMeis1* has a broad requirement for eye regeneration and the re-establishment of posterior poles by promoting the differentiation of eye progenitor cells and inducing the expression of *Djwnt1*, respectively, which further reveal the function of Meis1 in tissue identity determination during the regeneration process.

## 4. Materials and Methods

### 4.1. Species and Culture Conditions

Animals used in all experiments were a clonal strain of the planarian *Dugesia japonica* and were fed in autoclaved stream water at 20 °C [58]. Before all the experiments, animals were starved for at least 1 week [59], and those animals with a total of 5–8 mm in length were used for all experiments.

### 4.2. Gene Cloning

The ORF of *DjMeis1*, *DjMeis2*, *DjMeis3*, *β-catenin*, *Djovo*, *Djsix1/2*, and *Djeya* were identified in the planarian transcriptomic data [31]. Total RNA was extracted from 5 adult planarians by TRIzol (Vazyme, Nanjing, China). cDNA was synthesized from 1 μg of total RNA using Hiscript II reverse Transcriptase (Vazyme, Nanjing, China) and HiScript qRT Supermix II (Vazyme, Nanjing, China) by reverse transcription PCR. Sets of specific primers were designed to amplify the *DjMeis1* sequence from cDNA by PCR. All primers used to clone and synthesize dsRNA are shown in Appendix A.

### 4.3. RNAi

All the dsRNA (*DjMeis1*, *DjMeis2*, *DjMeis3*, *β-catenin*, *Djovo*, *Djsix1/2*, *Djeya*) used for the RNAi experiments were synthesized by in vitro transcription as previously described [31,60]. Briefly, T7 polymerase was used to synthesize dsRNA via in vitro transcription. DsRNA was denatured at 68 °C and annealed at 37 °C. Finally, dsRNA was extracted by ethanol precipitation. Next, dsRNA of each gene was diluted to 2 μg/μL and was injected to animals in the dorsal side one time per day for 1 week continuously using a Drummond microinjector. Then, 100 nL of dsRNA was injected to each animal every day. In double RNAi experiments, *β-catenin* and *DjMeis1* maintained a concentration of 2 μg/μL dsRNA. The water treated by DEPC was injected to the control group animals. Head, trunk, and tail fragments were amputated from the animals from the anterior and posterior sites of the pharynx at 24 h after the last injection.

### 4.4. Whole-Mount In Situ Hybridization

Whole-mount ISH (WISH) was performed as previously described [61]. In each experiment, ten animals were used in each group. Animals were killed in PBS (phosphate buffered saline) with 5% NAC (N-acetylcysteine) for 5 min and fixed in 4% paraformaldehyde for 30 min at room temperature. After dehydration in a methanol dilution series in PBST (0.3% Triton X-100 in PBS), animals were bleached in methanol with 6% H_2_O_2_ overnight under bright light. After rehydration, animals were treated by Proteinase K (20 mg/mL in PBST) for 10 min at 37 °C and fixed in 4% paraformaldehyde for 20 min at room temperature. Then, the animals were hybridized with DIG-labeled probes at 56 °C for 16-17 h and washed in 2× SSC (Saline Sodium Citrate buffer; Solarbio, Beijing, China) and 0.2× SSC three times for 20 min, respectively. Antibody incubation (1:4000; Anti–Digoxigenin-AP, Roche, Basel, Switzerland) and colorimetric (NBT/BCIP) were used for in situ detection. Different concentrations of SSC were prepared by dissolving 20× SSC in deionized water.

### 4.5. Whole-Mount Immunostaining

Whole-mount immunostaining was performed as previously described [58]. In each experiment, ten animals were used in each group. Animals were killed in PBS with 5% NAC for 5 min and washed three times with PBST at room temperature. Then, the animals were fixed in 4% paraformaldehyde for 2–4 h at 4 °C and incubated in 100% methanol for 1 h at −20 °C. Thereafter, the animals were blocked with 10% goat serum in PBST for 2–4 h at 4 °C and incubated with primary anti-synapsin (1:100; Developmental Studies Hybridoma Bank, Shanghai, China) or anti-H3p (1:250; Millipore, 05-817R, MA, USA) antibodies overnight at 4 °C. After six times of washing with PBST, the animals were labeled with goat anti-mouse Alexa Fluor 488 (1:500; Invitrogen, 673781, Shanghai, China) or goat anti-rabbit Alexa Fluor 568 (1:500; Invitrogen, 11036). Finally, the animals were observed by NIS element software (version 4.2.0, Olympus, IX73P1F, Tokyo, Japan).

### 4.6. Eye Resection

Eye resection was performed as previously described [62]. In each experiment, six animals were used in each group. Animals were placed on moist filter paper on a cold block in order to limit movement, while adjusting the focus and magnification of the dissector to make the eyes clearly visible. A microsurgery blade was used to remove the eyes through a small longitudinal dorsal incision.

### 4.7. BrdU Labeling

BrdU was performed as previously described [63]. In each experiment, ten animals were used in each group. Animals were treated with 1× Montjuic salts with 0.0625% N-acetylcysteine for 30–60 s three times and washed in 1× Montjuic salts for 1 min. Then, the animals were incubated in 1× Montjuic salts with 5 mg/mL BrdU (Sigma, Shanghai, China) for 1–2 h in the dark at 21 °C. After maintenance in 1× Montjuic salts at room temperature for 6–10 h, the animals were killed in 5% NAC for 5 min and fixed in 4% paraformaldehyde for 30 min at room temperature. Thereafter, 6% hydrogen peroxide in methanol was used to bleach the animals under bright light overnight. Next, the animals were rehydrated through a methanol dilution series in PBST and were treated with 2N HCl at room temperature for 45 min. After washing three times with PBST and blocking in PBST with 0.25% BSA at room temperature for 6 h, the animals were incubated in 1:1000 rat anti-BrdU (Proteintech, Beijing, China) overnight and washed in PBST eight times over 6 h the next day. Moreover, 1:500 goat anti-rat conjugated to HRP (Sangon Biotech, Shanghai, China) was used to label rat anti-BrdU at room temperature overnight. Finally, the animals were treated with tyramide conjugated to Alexa568 (Molecular Probes) for 30 min and were observed by NIS element software (version 4.2.0, Olympus, IX73P1F, Tokyo, Japan).

### 4.8. Fluorescence In Situ Hybridization

Fluorescence in situ hybridization was performed as previously described [61]. In each experiment, ten animals were used in each group. Briefly, animals were killed in 5% NAC for 5 min and fixed in 4% paraformaldehyde for 30 min at room temperature. After dehydration in a methanol dilution series in PBST, the animals were maintained in 100% methanol overnight at −20 °C. Then, the animals were rehydrated in 50% methanol for 10 min and washed in PBST and 1× SSC for 5 min, respectively. Formamide-bleaching solution (5% non-deionized formamide, 0.5× SSC, and 1.2% H_2_O_2_ in deionized water) was used to bleach the animals under bright light for 2 h. After washing in 1× SSC and PBST for 5 min and incubating in Proteinase K (20 mg/mL in PBST) for 10 min at 37 °C, the animals were fixed in 4% paraformaldehyde for 20 min and washed in PBST for 10 min. Then, the animals were hybridized with DIG-labeled PC2 probe at 56 °C for 16–17 h and washed in 2× SSC and 0.2× SSC three times for 20 min, respectively. PBST with 5% goat serum and 5% Western Blocking Reagent (Roche, Basel, Switzerland) was used to block the animals for 5 h. Next, the animals were incubated in 1:500 anti-Digoxigenin-POD (Roche, Basel, Switzerland) overnight. Finally, the animals were treated with tyramide conjugated to Alexa568 (Molecular Probes, Shanghai, China) and were observed by NIS element software (version 4.2.0, Olympus, IX73P1F, Tokyo, Japan). Different concentrations of SSC were prepared by dissolving 20× SSC in deionized water.

### 4.9. Statistical Analyses

Data were shown as means ± SD, and statistical analyses were performed by students. One-way analysis of variance (ANOVA) was used to analyze the data of two groups. A statistically significant difference was defined as *p* < 0.05.

## Figures and Tables

**Figure 1 ijms-24-03505-f001:**
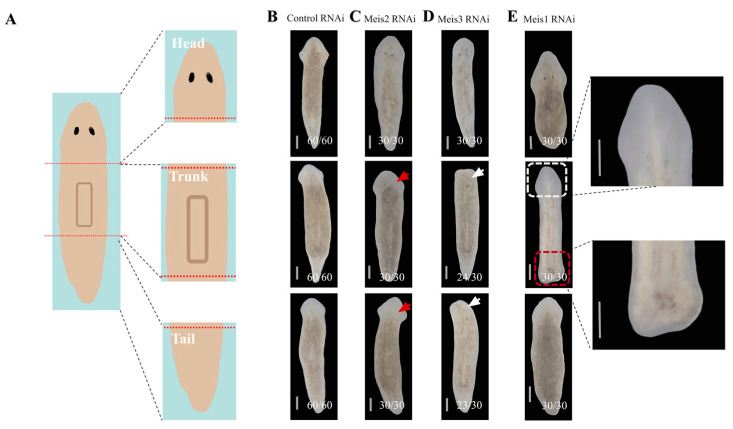
The regeneration phenotypes of planarians after knockdown of *Meis* family genes. (**A**) Planarians were amputated into three fragments from the anterior and posterior sites of pharynx at 24 h after the last injection. (**B**) The head, trunk, and tail fragments of control planarians developed into normal and complete individuals at 7 dpa. (**C**) The newly regenerated eyes (red arrows) in trunk (30/30) and tail (30/30) fragments of *DjMeis2* RNAi planarians were smaller than those in control groups at 7 dpa. (**D**) The trunk (24/30) and tail (23/30) fragments of *DjMeis3* RNAi planarians regenerated a squared head with elongated eyes (white arrows) at 7 dpa. (**E**) The head (30/30) and trunk fragments (30/30) of *DjMeis1* RNAi planarians failed to regenerate tails (red boxes). The trunk (30/30) and tail fragments (30/30) regenerated eyeless heads (white boxes) at 7 dpa. The images on the right were an enlarged view of the fragments in white and red boxes. Scale bars: 400 μm.

**Figure 2 ijms-24-03505-f002:**
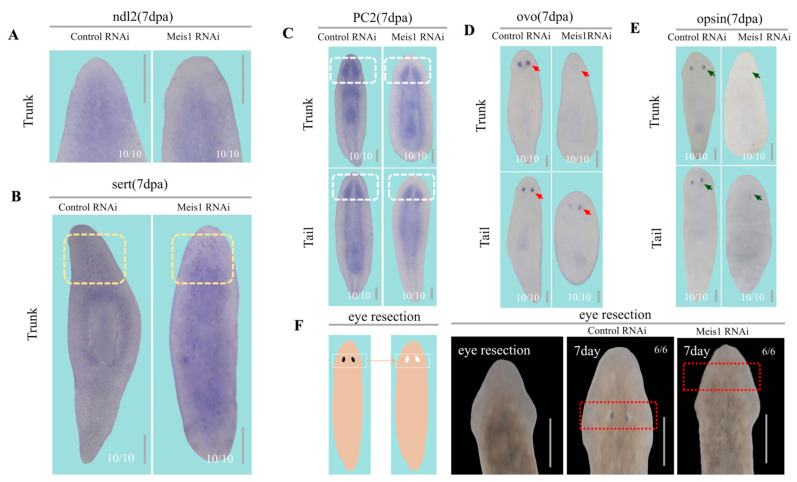
The effect of *DjMeis1* RNAi on the anterior regeneration. (**A**) The expression pattern of *ndl2* in trunk fragments by WISH. Only pre-pharyngeal regions were displayed here. *DjMesi1* RNAi planarians (10/10) normally expressed *ndl2* in the pre-pharyngeal regions. (**B**) Whole-mount ISH for *sert* in trunk fragments. *DjMeis1* RNAi planarians (10/10) normally expressed *sert* in the newly regenerated head (yellow boxes). (**C**) The images of CNS (labeled with *PC2*) in trunk and tail fragments. *DjMeis1* RNAi planarians (10/10) regenerated a proper bilobed brain (white boxes) in the newly regenerated head. (**D**) The regeneration of eye progenitor cells (labeled with *Djovo*) in trunk and tail fragments. *DjMeis1* RNAi planarians (10/10) regenerated fewer numbers of eye progenitor cells (red arrows) than the control groups. (**E**) The regeneration of mature eye cells (labeled with *opsin*) in trunk and tail fragments. *DjMeis1* RNAi planarians (10/10) did not express the mature eye cells marker, *opsin,* in anterior blastema (green arrows). (**F**) *DjMeis1* RNAi planarians (6/6) failed to regenerate eyes (red boxes) at 7 days after eye resection. Scale bars: 400 μm.

**Figure 3 ijms-24-03505-f003:**

*DjMeis1* RNAi had no effect on wound-healing and cell proliferation. (**A**) The head (10/10), trunk (10/10), and tail (10/10) fragments of *DjMeis1* RNAi planarians normally expressed the boundary marker *LaminB*, as indicated by WISH. (**B**,**C**) Phospho-H3 staining and quantitative statistical analysis of mitotic cells in planarian fragments. (**B**) *DjMeis1* RNAi planarians (10/10) retained normal proliferative ability at 6 h after amputation. (**C**) *DjMeis1* RNAi planarians (10/10) showed normal mitotic density compared to the control groups at 48 h after amputation. (**D**,**E**) Bromodeoxyuridine labeling experiments and quantitative statistical analysis of mitotic cells in the partial trunk fragments. (**D**) The anterior regions of pharynx (black box) of *DjMeis1* RNAi planarians (10/10) displayed the same level of proliferative ability compared to the control groups at 6 h after amputation. (**E**) The number of mitotic cells in the posterior wound site (black box) of *DjMeis1* RNAi planarians (10/10) had no significant difference with the control groups at 48 h after amputation. Statistical comparisons were conducted using the ANOVA test. Significant difference was defined as *p* < 0.05. ns *p* > 0.05. Scale bars: 400 μm.

**Figure 4 ijms-24-03505-f004:**
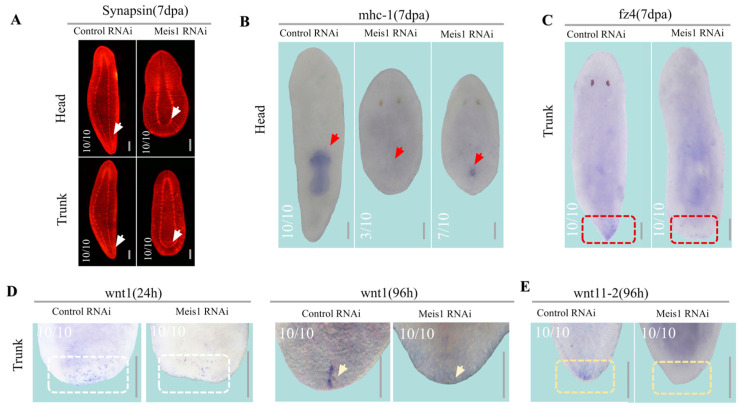
*DjMeis1* RNAi inhibited the reconstruction of posterior poles. (**A**) The images of CNS (stained with anti-synapsin) in trunk and tail fragments. *DjMeis1* RNAi planarians (10/10) did not regenerate VNC posteriorly (white arrows). (**B**) *DjMeis1* RNAi planarians did not regenerate (3/10) pharynx or regenerated incomplete pharynx (7/10), as indicated by *mhc-1* WISH (red arrows) in head fragments. (**C**) Whole-mount ISH for *fz4* in trunk fragments. *DjMeis1* RNAi planarians (10/10) did not express *fz4* (red boxes) in posterior blastema. (**D**) Whole-mount ISH for *Djwnt1* in newly regenerated posterior blastema of trunk fragments. *DjMeis1* RNAi suppressed the *Djwnt1* expression at both 24 and 96 h after amputation (white boxes and yellow arrows). (**E**) The expression pattern of *Djwnt11-2* in newly regenerated posterior blastema of trunk fragments by WISH. *DjMeis1* RNAi planarians failed to express *Djwnt11-2* at 96 h after amputation (yellow boxes). Scale bars: 400 μm.

**Figure 5 ijms-24-03505-f005:**
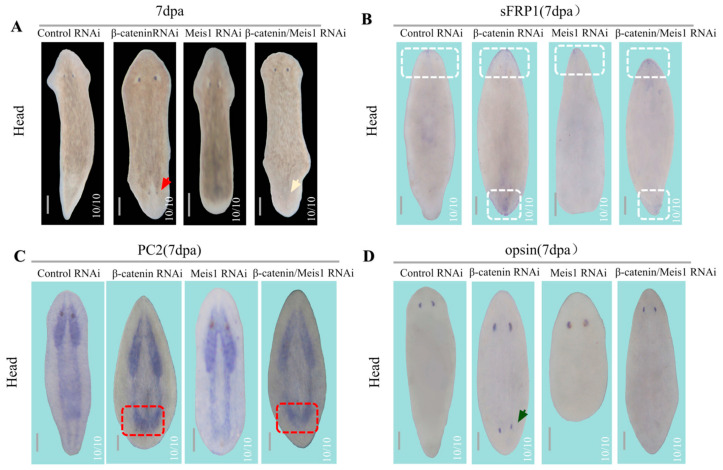
*β-catenin/DjMeis1* RNAi caused anteriorization of posterior poles in planarians. (**A**) Head fragments of *β-catenin* RNAi planarians (10/10) regenerated another head with a pair of eyes (red arrows) from posterior blastema. *β-catenin/DjMeis1* RNAi planarians (10/10) regenerated eyeless heads (yellow arrows) from posterior blastema. (**B**) The expression pattern of *sFRP1* (white boxes) in head fragments by WISH. *sFRP1* was observed in both anterior and posterior blastema of *β-catenin* RNAi (10/10) and *β-catenin*/*DjMeis1* RNAi planarians (10/10). (**C**) The images of CNS (labeled with *PC2*) in head fragments. Another pair of brain nerves (red boxes) were observed in posterior blastema of *DjMeis1*/*β-catenin* RNAi and *β-catenin* RNAi (10/10) planarians. (**D**) The regeneration of mature eye cells (labeled with *opsin*) in head fragments. Only *β-catenin* RNAi planarians (10/10) expressed *opsin* (green arrows) in posterior blastema. Scale bars: 400 μm.

**Figure 6 ijms-24-03505-f006:**
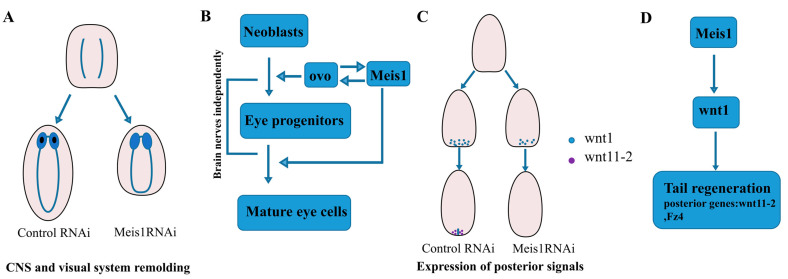
Summary of *DjMeis1* functions on regeneration. (**A**) *DjMeis1* is required for the proper reconstruction of VNC and eye regeneration. (**B**) The model of Meis1 regulates the eye regeneration. (**C**) *DjMeis1* is required for the *Djwnt1* and *Djwnt11-2* expression. (**D**) The model of Meis1 functions on posterior regeneration.

## Data Availability

Not applicable.

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
