# Peer review of "Meis1 Controls the Differentiation of Eye Progenitor Cells and the Formation of Posterior Poles during Planarian Regeneration"

_ijms, 2023, doi:10.3390/ijms24043505_

Round 1

Reviewer 1 Report

In this manuscript, Wang and colleagues described a role of the transcription factor Meis1 in the regulation of eye and posterior structure formation in the regenerating planarian Dugesia japonica. By extensive RNAi experiments, the auhors demonstrate that DjMeis1 plays a role, in addition to DjMeis2 and DjMeis3 in eye formation, with DjMeis1 probably impacting early neoblast differentiation processes during head regeneration. In addition, DjMeis1 RNAi disrupt posterior structures whose formation is controlled by the Wnt/b-catenin pathway. In this process, DjMeis1 seems to be involved in differentiation rather than proliferation of neoblasts, as assessed through H3S10 phoshorylation. Overall, this manuscript describes interesting results but suffers from a lack of integration with what is known from the literature about the function of Meis proteins and their Pbx partner in neoblast differentiation. Indeed, it is not clear what degree of novelty is brought by the present work and the authors should put some efforts in highlighting novel findings. Below are a number of suggestions that could improve the manuscript.

1. The manuscript suffers from a large number of misphrased sentences. It should be edited thoroughly.

2. Although it is interesting to compare Meis1,2 and 3 sequences between Dugesia japonica and Homo sapiens or Mus musculus (Fig S1), it would be quite informative also to show alignments between Meis proteins from Dugesia japonica and Schmidtea mediterranea. In particular, this would allow to ascertain that DjMeis3 indeed corresponds to the so-called SmedMeis-like gene. In this respect, the absence of eye in trunk regeneration experiment under SmedMeis-like RNAi (Rodriguez-Esteban et al, BMC Genomics, 2015) is somehow in contradiction with the milder phenotype observed here in DjMeis3 knock-down (KD) animals. In addition, the authors should compare their results with those from Blassberg et al (Development, 2013) describing the phenotype of SmedPbx KD animals.

2. In their study of eye regeneration in Smed, Lapan and Reddien (Cell Reports, 2012) showed that SmedMeis (equivalent to DjMeis2) KD impacts late eye morphogenetic events but not proliferation and differentiation of progenitors expressing Ovo. Interestingly, the phenotype described here for DjMeis1 is more similar to the phenotype described for SmedOvo, suggesting that DjMeis1 could play a role in progenitors. Consistent with this hypothesis, the authors found that expression of Ovo is reduced in DjMeis1 KD conditions (Fig. 2). This should be explored further through additional KD experiments combined with WISH. For example, is knocking-down Ovo affecting DjMeis1 expression? This should be done also for Eya and Six1/2 which are important transcription factors co-expressed with Ovo.

3. In the data shown in Fig 3., I find the H3p images not so demonstrative. Could the authors use another method like EdU incorporation to investigate proliferation?

4. The data shown in Fig. 4 could be nicely complemented by running Dil injection to mark nerve cords (note that VNC -line 227- is not defined in the article).

5. In Figure 5, I do not understand why no eyes are visible in the Meis1 KD condition in panel D. Since this is showing posterior regeneration from a head fragment, one would expect to observe eyes. The authors should comment on that or describe the set up in a more comprehensible manner.

6. In line 54 of the introduction, Djpbx and Djislet are mentioned. Are they position control genes (PCGs) as described line 55? The authors should describe the function of PCGs in more detail in the introduction. Line 142, the authors give the abbreviation of PCGs but it has already been given line 55.

7. The authors should add a diagram summarizing their findings about the roles of DjMeis1 in the establishment of the antero-posterior (AP) axis in the planarian in relation with Wnt signaling.

Reviewer 2 Report

The manuscript by Wang et al. represents interesting experimental research on planarian regeneration. In particular, authors experimentally confirm the involvement of Meis1 gene in differentiation of eyes progenitor cells and the formation of posterior poles through control of Wnt signaling during regeneration in planarians. The study is based on sufficient material and utilize broad range of methods: two types of surgical operations, RNAi, whole-mount in situ hybridization and immunostaining. Results of the study represent original material and will be interesting for a broad scientific auditory. However, I have some considerations about the manuscript and recommend publishing it in International Journal of Molecular Sciences after a minor revision. All my considerations concern only representation of the data, but not the scientific results per se.

General

1. The main drawback of the manuscript is the representation of Materials and Methods. Currently, this section of the manuscript is too short and superficial. All additional important information about methods and experiments can be inferred from Results section, but it hiders the understanding of experimental procedures and makes it almost impossible to repeat the study. Materials and Methods section should be thoroughly rewritten and expanded. I put particular considerations about this section in following parts of the review.

2. The English language should be considerably improved in the whole manuscript. Some sentences are unclear and vague, while others are grammatically incorrect. I put few examples in the following review, but I could miss some others, so the whole manuscript should be thoroughly checked and improved.

Abstract

1. Line 16. Please, add that Djwnt1 is a ligand of Wnt signaling cascade. It will make your point clearer for readers.

Introduction

1. Line 27 “…embryonic cortical by promoting neuronal…” – I believe there is a missing word between “cortical” and “by”.

2. Line 30 “…the formation of cardiac[3]…” – I believe there is a missing word after “cardiac”.

3. Line 32. Remove “On the other hand”. No need of it here. Also, it is repeated in the next sentence.

4. Line 38-39. For me, it is unclear how your introduction goes from embryonic roles of Meis1 to the question of the role of this protein for stem cell fate. Maybe you should slightly rewrite the last sentence.

5. Line 55-56. The sentence “These studies reveal…” is unclear for me. Please, rewrite.

6. Line 58-61. I recommend writing these sentences in Present tense.

Materials and methods

Major considerations:

1. Line 73-79. What genes were studied in RNAi experiments? In the first sentence of this paragraph, you write only about DjMeis1. But in the third sentence there is “In double RNAi experiments, each gene kept…”. The phrase “each gene” implies that there were more than one gene (in the Result section I found information about at least two other genes, DjMeis2 and DjMeis3, used in RNAi experiments). Please, specify information about genes clearly in Material and method section!

2. Line 78-79. You write that operations were done “at 24 h after the last injecting”. How many injections were done to each animal before operation? At what period? By dsRNA of what gene? Please, specify this information clearly for each particular experiment!

3. Please, provide extensive information about replications of RNAi experiments, number of specimens per RNAi experiment, number of specimens used for WMISH and immunostainings. I found some of this information in the Result section, but it should be obligatory represented in Material and Methods section of the manuscript. Possibly it will be useful to represent this information as one or several tables.

Minor considerations:

1. Line 67-68. The sentence “Stop feeding…” is grammatically incorrect. Please, rewrite.

2. Line 70-71. The sentence “Used a pair…” is grammatically incorrect. Please, rewrite.

3. Line 92-93. The sentence “Then used PBST with…” is grammatically incorrect. Please, rewrite.

4. Line 93-94. The sentence “Then blocked samples…” is grammatically incorrect. Please, rewrite.

5. Line 98-99. Please, specify the microscope model used for study of the whole-mount immunostainings in the corresponding subsection.

6. Line 105. I believe the students who performed statistical analysis should be included in the Acknowledgments of the manuscript.

Results

Major considerations:

1. Please, unify the representation of the time after the amputation: currently you are using hours after amputation (hpa) in the main text, but days after amputation (dpa) in figures and their captions.

2. In the figures captions, you provide information about the number of specimen with a particular phenotype after each of the experiment (for example, “The newly regenerated eyes (red arrows) in trunk (30/30) and tail (30/30) fragments”). Please, also provide this information in the same format in the main text of the Results for each experiment. It will make your descriptions more ease for quick analysis by readers.

3. Line 120-129. I recommend describing DjMeis1 RNAi phenotypes more accurately and consistently, as it is a central part of the manuscript. It should be absolutely clear for readers. I think you should start with the description of the eyeless phenotype (and its comparison with DjMeis2 RNAi and DjMeis3 RNAi phenotypes), then proceed with the failure of tail regeneration, and finally present the eyeless and tailless phenotype after trunk regeneration.

4. Figure 2A, B. Why there are no eyes in control specimens? In the section 3.1 you stated that control specimens regenerate normal individuals within 7 dpa.

5. Figure 2D. What is that staining in the central body parts of all specimens (it is especially obvious in the control specimen)? Are cells in this region express ovo or is it some unspecific binding?

6. Line 260-261. You stated about DjWnt1 expression at 1 dpa that “…DjMeis1 RNAi animals displayed a similar expression patterning compared to control groups on 24h after amputation…”. However, the staining of the control specimen in Figure 4D (1 dpa) is much more prominent in my opinion. If this staining in control specimen represent unspecific binding, I recommend changing this image with more illustrative for your statement. However, if this (more extensive) staining is due to (more extensive) expression of DjWnt1, the description of the results (as well as your conclusion based on these results) should be changed and reworked.

7. Line 261-262. Here is somewhat similar situation: here you stated that DjMeis1 RNAi specimen at 4 dpa show no expression of DjWnt1 comparing to control specimens. But according to Figure 4D (4dpa), the control specimen also contains a neglected amount of cells expressing DjWnt1 on its posterior pole. Could such a minor differences qualified as significant? Similarly to the previous consideration, I recommend changing the image of control specimen with more illustrative or change the description of the results.

8. Figure 4D (4 dpa). What is that extensive staining in more anterior parts of control specimen? Are cells in this region express DjWnt1 (if so, is this expression important for axis patterning?) or is it some unspecific binding?

Minor considerations:

1. Line 131. The title of the Figure 1 should be changed, as this figure shows not only the effect of DjMeis1 RNAi on the planarian regeneration, but also the effect of two other genes.

2. Line 151. The title of the Figure 2 should be changed as well, as this figure also shows more information than the current title implies.

3. Line 151. Here and in all other cases, I recommend using the phrase “expression pattern” instead of “expression patterning”.

4. Line 214. The title of the Figure 3 should be changed, as this figure shows not only the effect of DjMeis1 RNAi on cell proliferation, but also on LaminB expression.

5. Line 227. What is VNC? Please provide the full phrase for this abbreviation here.

6. Lime 230. Please, specify clearly here what do you mean using the phrase “…the regeneration of the posterior organ”. Currently, the sentence is vague.

7. Line 271. The title of the section 3.5. contains mistyping.

8. Line 283. I recommend changing the phrase “…caused posterior poles of planarian anteriorization…” with the phrase “…caused anteriorization of posterior poles in regenerating planarian…”

Discussion

1. The fragments Line 320-330 and Line 345-351 represent the repeats of the Results. While they are somehow important for discussion, they should be shortened as much as possible.

Round 2

Reviewer 1 Report

The authors have now significantly improved their manuscript and addressed all my concerns.